# Purification and Partial Characterization of a Bacteriocin Produced by *Lactobacillus pentosus* 124-2 Isolated from Dadih

Tri Yuliana *, Annisa Rizka Pratiwi, Shafa Zahratunnisa, Tita Rialita, Yana Cahyana, Putri Widyanti Harlina and Herlina Marta

Faculty of Agro-Industrial Technology, Universitas Padjadjaran, Jl. Raya Bandung Sumedang KM 21, Sumedang 45363, Indonesia
* Correspondence: t.yuliana@unpad.ac.id; Tel.: +62-22-7798844

**Abstract:** The fermentation process results in the presence of lactic acid bacteria (LAB). Traditional fermented buffalo milk from the Indonesian province of West Sumatra is known as dadih. Bacteriocins are naturally occurring antimicrobial peptides that can be produced by LAB. These bacteriocins have the potential to be used as natural preservatives. This research was conducted with the intention of isolating and partially characterizing a bacteriocin that was generated by *Lactobacillus pentosus* 124-2 that was isolated from dadih. Using MRS agar media and 16sRNA, the LAB that was found in dadih was successfully isolated. Bacteriocins were produced using media consisting of MRS broth. After the bacteriocins were obtained, they were subjected to a series of purification processes, one of which included ammonium sulfate (70%) followed by gel filtration chromatography for additional purification. According to the findings, the strain of LAB that was discovered in the dadih was *Lactobacillus pentosus* 124-2. The specific activity of the bacteriocins rose, allowing for the effective completion of the purification process. The pure bacteriocins had the greatest specific activity values, at 60.59 AU/mg, while the yield values were 0.41% and 3.15-fold. Bacteriocins in their purest form were able to inhibit the growth of Salmonella species as well as Staphylococcus aureus. The characterization results included partial bacteriocins that were resistant to pH 2, 4, and 6; antibacterial activity that was was stable at a temperature range of 25 °C to 121 °C; and resistance to the addition of 2%, 4%, and 6% salt concentrations. Microbial stability against the addition of surfactants EDTA, SDS, and Tween 80 was also obtained. In light of these findings, a bacteriocin derived from *L. pentosus* possesses the possible ability to be utilized in the food business as a biopreservative.

**Keywords:** bacteriocin; dadih; purification; *Lactobacillus pentosus*; biopreservative

## 1. Introduction

Foodborne illnesses are caused by consuming food that is contaminated by the presence of bacteria, viruses, or parasites that have entered the body [1]. Preservatives are used for heating and drying, but they can reduce the quality and change the texture of the food [2]. The shelf life of foods is currently being extended by the use of preservative chemicals, many of which are dangerous. Such chemicals are especially dangerous in non-food groups. In addition, some people think that these chemicals are not good for health because they can cause diseases, such as cancer. These drawbacks have triggered the discovery and development of natural preservatives [3].

A biopreservative involves the use of microorganisms or natural products [3]. Biopreservative substances such as lactic acid bacteria (LAB), which have antibacterial properties, have long been used [1]. However, during the process of using biopreservatives, changes in the taste, texture, and odor of a product take place, so the use of lactic acid bacteria as a biological preservative has been limited. It can safely extend the shelf life of food without causing health effects and it can produce other compounds with antibacterial

properties, specifically other bacteria, so research regarding the isolation and potency of LAB is currently underway.

Bacteriocins are antibacterial peptides that have received GRAS (generally recognized as safe) status from the FDA, so it is safe to use them as biological preservatives [4]. Bacteriocins, as biological preservatives, can be used to control microbial contaminants in foods, but their commercial potential is limited and their costs are relatively high. Bacterial production in Indonesia requires much more exploration and innovation. Among the efforts undertaken to explore the production of bacteria is the production of bacteria from food. Bacteria-producing LAB, such as *Lactobacillus pentosus*, have been found in some fermented foods. The bacterium *L. pentosus* can be isolated from fermented products such as dadih, which is made from the milk of buffalo that are native to West Sumatra, Indonesia [5].

The bacterium *L. pentosus* was isolated from the feces of healthy infants and showed potent antibacterial activity and probiotic compounds [6]. The newly studied dadih bacteriocin *L. pentosus* is a crude bacteriocin, showing antibacterial activity against *Salmonella* sp., *Escherichia coli*, and *Staphylococcus aureus* [7].

A purification process can increase the specific activity of bacteriocins, so that the resulting activity is purely that of bacteriocins and is safer for consumption because it reduces other components that are not safe to eat or are of non-food grade [8,9]. In addition, to determine the activity of pure bacteriocin, it is necessary to test the antimicrobial activity against indicator bacteria [8,10]. A partial bacteriocin characterization was carried out to determine the characteristics of partial bacteriocins through testing on pH, temperature, surfactant, and salt. A food test can predict the properties of bacteriocins at the level of partial purification for application. Exploration and purification of bacteriocins from food will always present new alternatives, so characterization is carried out.

In this study, *L. pentosus* was isolated from fermented buffalo milk. The bacteriocin was produced in three steps: the production of crude bacteriocin, the partial purification by ammonium sulfate precipitation, and the purification by gel filtration chromatography with Sephadex LH-20. The antibacterial activity, specific activity, and molecular weight of bacteriocins were determined. In addition, the effects of temperature, pH, surfactants, and salt were also considered to determine the stability of bacteriocin.

## 2. Materials and Methods

### 2.1. Bacterial Indicator Strains and Media

*Salmonella* sp. and *Staphylococcus aureus* were used as the indicator strains. They were grown in NA medium at 37 °C.

### 2.2. Isolation and Screening of Bacteriocin-Producing Strain

The isolation of bacteriocin candidates was carried out in several stages, including the activation of LAB isolates, the isolation of bacteriocins from LAB culture, and purification. The isolation process uses a dilution method and is modified in the multilevel dilution section [11]. In this study, the dilution process was carried out by adding 1 g of a dadih sample into a test tube containing 9 mL of 0.85% NaCl solution. The solution was homogenized by the vortex. Bacterial isolation was carried out using the enrichment method, in which 50 μL of the dilution results from $10^{-3}$ to $10^{-8}$ were taken and inoculated into sterile Petri dishes containing MRS agar media which was added with 1% $CaCO_3$ in duplicate. The isolates were incubated at 37 °C for 48 h to obtain LAB colonies.

### 2.3. Identification of LABs by Sequencing of 16S-rDNA

The isolate with the highest antibacterial activity against *Salmonella* sp. and *Staphylococcus aureus* was also identified using 16S rRNA sequence analysis. This DNA was used as a template to amplify the 16S rRNA gene through polymerase chain reaction (PCR) using universal primers (27F: 5′-AGA GTT TGA TCM TGG CTC AG-3′ and 1492R 5′ -TAC GGY TAC CTT GTT ACG ACT T-3′). Homology searches against NCBI databases were con-

ducted using the BLASTN program. Phylogenetic analysis was performed using Mega11 (MEGA Software, Philadelphia, PA, USA) with the neighbor-joining method.

### 2.4. Production of Crude Bacteriocin

Crude bacteriocin production was carried out by the reactivation of isolates, which were refreshed on MRSB media to obtain cell-free supernatant (CFS). LAB culture isolates (10%) were grown in MRSB media and incubated for 18 h at 37 °C. LAB liquid cultures were centrifuged at 6000 rpm for 10 min at 4 °C, followed by sterile filtration using a millipore 0.22 μm PVDF. Cell-free supernatant was neutralized by adding 1 N NaOH to pH 6.5 [10].

### 2.5. Purification of Bacteriocin

The cell-free supernatant was then precipitated with ammonium sulfate (70%) at 4 °C [12]. The mixture was separated at 6000 rpm for 1 h using centrifugation. The product was obtained as a pellet. The pellet obtained was then separated and dissolved in a phosphate buffer. The dissolved pellets were then purified by gel filtration chromatography using a column measuring $2.5 \times 50$ cm (Sephadex LH-20; Cytiva, Uppsala, Sweden) as a steady phase and 80% of methanol and 20% of water as a mobile phase. Sephadex LH-20 was weighed and added with 80% of methanol and 20% of water, then stirred slowly and put into the column. The eluent was slowly added and collected in a beaker glass until the Sephadex expanded and solidified. After that, the bacteriocin sample was added to the column through the column walls while eluting with solvent. The eluent was collected in a vial and marked with the first fraction. The eluent was collected every 1 mL until the last clear eluent was reached. The active fraction was collected in vials every 1 mL and tested by the paper disc diffusion method to get the best fraction [13]. The best fraction was tested for its protein content and its molecular weight was identified.

### 2.6. Antibacterial Activity of Bacteriocin

The antibacterial activity of bacteriocin against indicator bacteria was indicated by the formation of a clear zone around the disc paper; it was expressed in arbitrary units per mL (AU/mL) [13]. The total bacteriocin activity (AU) was obtained from the product of the bacteriocin activity ($mm^2$/mL) and the sample volume (mL). Total protein was obtained from the product of the sample volume (mL) and the protein concentration measured by the Bradford method (mg/mL). Specific activity was calculated by dividing the total bacteriocin activity (AU) by the total protein (mg). Each stage of purification had a different recovery value obtained from the difference between the total activity after purification and the total activity before purification, multiplied by 100 [14]. The level of purification of bacteriocins was obtained from the difference between the specific activity after purification and the specific activity before purification [15].

### 2.7. Determination of Protein

The determination of protein content was carried out using the Bradford method. A protein standard curve was produced from bovine serum albumin (BSA), which was dissolved in distilled water in concentrations of 0 μg/mL, 2.5 μg/mL, 5 μg/mL, 7.5 μg/mL, 10 μg/mL, and 12.5 μg/mL; 0.1 mL of each concentration was then added to 5 mL of Bradford's reagent, homogenized, and incubated for 15 min at room temperature. The absorbance was measured using a spectrophotometer with a wavelength of 595 nm. Samples of crude bacteriocin, partial bacteriocin, and pure bacteriocin were measured for protein concentration by adding 5 mL of Bradford's reagent to 0.1 mL of sample, homogenized, and incubated for 15 min at room temperature. The absorbance was measured using a spectrophotometer at a wavelength of 595 nm. The absorbance was entered in a linear regression curve from the protein standard curve [16].

## 2.8. Determination of the Molecular Weight

The SDS-PAGE concentration of 12% determined the molecular weight of bacteriocin. Electrophoresis was carried out at 150 volts until the tracer dye reached the gel's bottom. After the electrophoresis process was completed, the gel was stained with Coomassie brilliant blue. Then, the stain was removed by washing for 24 h using a mixture of acetic acid, methyl alcohol, and water. Molecular weight was determined by SDS-PAGE protein molecular weight pre-stained markers [17].

## 2.9. Partial Characterization

Characterization was carried out on partial bacteriocins with resistance to different conditions including temperature, pH, salt, and surfactant. The bacteriocin activity was tested for temperature resistance in several treatments: 25 °C, 37 °C, 45 °C, 60 °C, and 80 °C for 30 min and a temperature of 121 °C for 15 min. pH variations were 2, 4, 6 and 9. Tests of resistance to salt concentrations were carried out by looking at the activity of bacteriocins in different concentrations of NaCl, which were 2%, 4%, and 6% concentrations. Then, the stability activity of bacteriocins was tested against the addition of surfactant, EDTA, SDS, and Tween 80 [9]. At the end of the test, bacteriocin stability was tested for antimicrobial activity using the disc method.

## 2.10. Antimicrobial Activity Test

Antimicrobial activity testing was carried out using the paper disc diffusion method or the Kirby-Bauer method. The solid MHA media in the petri-dish was given indicator bacteria that had been inoculated into NaCl 0.85%. The bacteriocin samples were dropped onto sterile 6 mm paper disks and then placed on solid MHA media containing indicator bacteria and incubated at 37 °C for 24 h. Antimicrobial activity was indicated by the formation of a clear zone around the disc which was then measured with a caliper in millimeters (mm) [18].

## 3. Results

### 3.1. Isolation and Screening of Bacteriocin-Producing Strain

At the stage of isolation of bacteriocin-producing bacteria from dadih can be seen in Table 1, there were five isolates that were identified by colony morphology: shape, border, color, elevation, and Gram-staining with 3% H2O2 and catalase test.

**Table 1.** Identification qualitative and morphology isolate.

| Isolate | Qualitative | | Morphology | | | | |
|---|---|---|---|---|---|---|---|
| | [a] Catalase Test | [b] Gram Stain | Cell Shape | Form | Edge | Elevation | Color |
| DSK 1 | (−) | (+) | Bacilli | Round | Slippery | Convex | Milky White |
| DSK 2 | (−) | (+) | Bacilli | Round | Slippery | Convex | Milky White |
| DSK 3 | (−) | (+) | Bacilli | Round | Slippery | Raised | Milky White |
| DSK 4 | (−) | (−) | Bacilli | Round | Slippery | Convex | Milky White |
| DSK 5 | (−) | (−) | Bacilli | Round | Slippery | Raised | Milky White |

[a] Positive results in the catalase test were indicated by the presence of air bubbles; negative catalase results (no gas bubbles) indicated that the isolates did not produce catalase enzymes that convert $H_2O_2$ into water and oxygen.
[b] Gram-positive bacteria appear as purple-blue and Gram-negative bacteria stain appear as pink-red.

Based on identification, qualitative and morphology showed that isolates DSK 1, DSK 2, and DSK 3 have LAB characteristics: a rod shape or a coccobacilli cell shape; catalase-negative; Gram-positive; non-spore; acid-tolerant; low G + C (guanine + cytosine); aero-tolerant or anaerobic; and able to ferment sugar into lactic acid [18].

### 3.2. Candidate for Bacteriocin Production

The isolation process showed that the qualitative identification of isolates DSK 1, DSK 2, and DSK 3 showed its LAB characteristics. A selection of candidate isolates

was carried out to determine which isolates would be used as bacteriocin producers. The selection process was performed by testing the antibacterial activity of liquid LAB isolates. The paper disc or Kirby–Bauer method was used as the test method. The Kirby–Bauer method was chosen because it is a rapid, simple, and accurate antimicrobial test method [19]. Antibacterial activity was determined by measuring the diameter of inhibition. The diameter of inhibition was measured after subtracting the disc diameter. The results of the antibacterial activity test are shown in Figure 1.

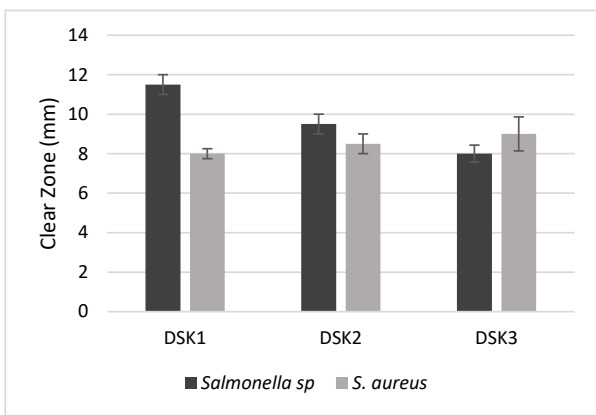

**Figure 1.** Antibacterial activity of bacteriocin-producing candidates.

Antibacterial activity testing showed that LAB isolates had antibacterial activity against Gram-negative and Gram-positive bacteria. The selection showed that the DSK1 isolates were the LAB isolates with the highest antibacterial activity, with 11.5 mm (*Salmonella* sp.) and 8 mm (*S. aureus*) inhibition zones.

### 3.3. LAB Identification by 16S-rDNA Sequencing

The identification method used was 16S rRNA analysis, which identifies bacteria at the species level based on base pair sequences [20]. Sequence analysis results showed that the lactic acid bacteria were identified as *Lactobacillus pentosus* 124-2. Identification was performed by amplification using primer pairs 785F and 907R. Sequencing results were processed using BioEdit software, then determined using the basic local alignment finder (BLAST) program from the National Center for Biotechnology Information. The DSK1 isolate identified as strain *L. pentosus* 124-2 with an e value of 0.0, a 98% query range, and a 99% identity value.

A phylogenetic analysis was performed to describe the relationships between species in a branch, such as a tree known as a phylogenetic tree. Phylogenetic analysis uses DNA or protein molecular data to describe evolutionary relationships between species [21]. A phylogenetic tree uses the neighbor join method of Molecular Evolutionary Genetic Analysis (MEGA) software, version 11.0. The analysis of the phylogenetic tree was statistically checked using the bootstrap method. The bootstrap value was used as a reference for the confidence level—the larger the obtained value, the greater the confidence [22].

Analysis showed (Figure 2) that *L. pentosus* 124-2 of dadih consisted of three inner groups and one outer group. The inner group was divided into three branches: clade I, clade II, and clade III. Group I included five species: *L. plantarum* NBRC 15891 (NR_113338.1), *L. plantarum* subsp. JCM 1149 (LC064896.1), *L. plantarum* JCM 1149 (NR_115605.1), *L. plantarum* CIP 103151 (NR_104573.1), and strain *L. paraplantarum* DSM 10667 (NR_025447.1). In clade II, the most closely related species were *L. plantarum* JCM 1149 (NR_117813.1), *L. pentosus* JCM 1558 (LC071808.1), and *L. plantarum* 15891 (NR_112690.1); clade III included a species of *L. plantarum* sp. 22 (NR_042254.1). This species was separate from the other *Lactobacillus* species but was related to *L. plantarum* JCM 1149 (NR_117813.1), with a bootstrap value of 64.

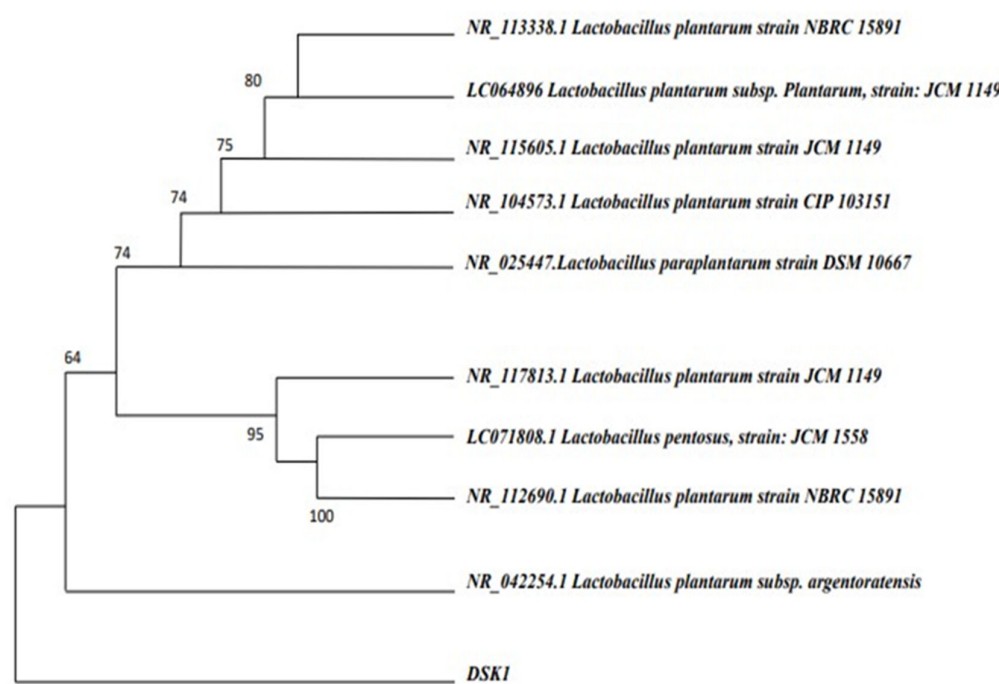

**Figure 2.** Phylogenetic tree of *L. pentosus* 124-2.

In the outer group, the known species was DSK 1 *L. pentosus* 124-2. The outer group is a group of living things that are considered to be ancestors or ancestors. Ancestor is a term for species that have undergone speciation events to give rise to offspring species [23]. The DSK 1 isolate is in the outer group. It could be concluded that the DSK 1 isolate was the ancestor of nine other species.

### 3.4. Purification of Bacteriocin

The resulting crude bacteriocin was treated with ammonium sulfate at a concentration of 70% to precipitate and pellet the peptide, and the pellet was dissolved in buffer and purified by Sephadex LH-20 gel filtration chromatography to prepare five fractions, coded F1, F2, F3, F4, and F5. The second fraction, code F2, had the highest level of antibacterial activity against *Salmonella* sp. The produced one was 14.375 ± 2.65 mm and *S. aureus* was 10.375 ± 1.95 mm (Figure 3).

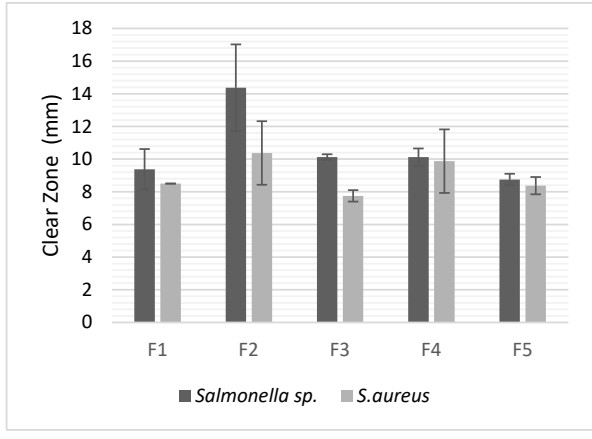

**Figure 3.** Antimicrobial activity of purified fraction.

After two purification steps—ammonium sulfate precipitation and gel filtration chromatography—the *L. pentosus* bacteriocin was purified 3.15-fold with a yield of 0.41%

(Table 2). The specific activity of the bacteriocin increased; the most significant value of specific activity was 60.593 AU/mg, which was produced by the bacteriocin at the final stage of purification using gel filtration chromatography. In a previous study, pentocin JL-1 was purified by gel filtration chromatography and showed a 25-fold yield value of 11% with a specific activity of 924 AU/mg [24].

**Table 2.** The antibacterial activity of bacteriocin against *Salmonella* sp. and *S. aureus*.

| Purification Stage | Volume (mL) | Protein Concentration (μg/mL) | Total Protein (mg) | [a] Total Activity (AU) | [b] Specific Activity (AU/mg) | [c] Yield (%) | [d] Purification (Fold) |
|---|---|---|---|---|---|---|---|
| Cell-Free Supernatant (CFS) | 1300 | 51.50 | 66.95 | 1609.91 | 24.05 | 100 | 1 |
| Precipitation Ammonium Sulfate | 38 | 106.63 | 4.052 | 78.06 | 19.27 | 4.85 | 0.80 |
| Gel Filtration Chromatography | 2 | 53.83 | 0.108 | 6.524 | 60.59 | 0.41 | 3.15 |

[a] Total activity (AU). Activity level (AU/mL) × volume (mL). [b] Specific activity (AU/mg) represents total activity divided by protein concentration. [c] Yield (%) represents total activity of purified fraction divided by total activity of crude extract × 100. [d] Purification fold represents specific activity of purified fraction divided by specific activity of crude extract activity (AU/mL) × volume (mL).

Bacteriocins purified by gel filtration chromatography were tested for antibacterial activity by the Kirby–Bauer method and compared to antibiotics (Table 3). In this study, pure bacteriocins were tested for their ability to inhibit the indicator bacteria *Salmonella* sp. and *S. aureus*.

**Table 3.** Antimicrobial activity of purified bacteriocin.

| Sample | Clear Zone (mm) | |
|---|---|---|
| | *Salmonella* sp. | *S. aureus* |
| Bacteriocin *L. pentosus* | 11.79 ± 2.51 | 9.96 ± 1.31 |
| Antibiotic | 30 | 18 |

Based on the results, pure bacteriocin formed zones of inhibition against *Salmonella* sp. of 11.79 ± 2.51 mm and against *Staphylococcus aureus* of 9.96 ± 1.31 mm. These results were lower than those of antibiotics, with a 30 mm zone of inhibition against *Salmonella* sp. and an 18 mm zone of inhibition against *S. aureus*.

### 3.5. Determination of the Molecular Weight of Bacteriocin L. pentosus

The molecular weight of the bacteriocin was confirmed with the partially purified bacteriocin, and the second fraction of bacteriocin purified by gel filtration chromatography showed the greatest antibacterial activity. Two bands of partially purified bacteriocin with molecular weights of 26.69 kDa and 17.15 kDa were produced, while the pure bacteriocin had a molecular weight of 17.15 kDa (Figure 4).

Further purification of partially purified bacteriocins was justified because distinct bands indicated that the protein produced was pure. This result differed from a previous study, in which the bacteriocin *L. pentosus* purified by pH adsorption and gel filtration chromatography using Sephadex G-10 had a molecular weight of 5.59 kDa [25]. This indicated that different purification methods may produce bacteriocins with different molecular weights and that any judgment that a bacteriocin is truly pure must be supported by a specific activity value.

### 3.6. Partial Characterization of the Isolated Antibacterial Substance

The antibacterial substance isolated from strain 124-2 was highly stable in response to heat. The effect of pH on antimicrobial activity showed that the antimicrobial retained 100% of its activity from pH 2 to 6 and decreased at pH 9 (Figure 5A). Based on these values, it can be concluded that the bacteriocin *L. pentosus* is constant at temperatures between

25 °C and 121 °C, with optimal conditions at temperatures between 25 °C and 60 °C. When applying the *L. pentosus*-derived bacteriocin, the recommended temperature for optimal bacteriocin efficacy was below 80 °C (Figure 5B). A bacteriocin activity test was performed to determine the tolerance of the bacteriocin to salinity. Tests were performed at different concentrations of NaCl: 2%, 4%, and 6%. The results showed that *L. pentosus* is tolerant to salt conditions (Figure 5C). The activity of the compounds increased after treatment with inhibitors such as SDS, Tween 80, and EDTA (Figure 5D). Relatively small size peptide molecules with uncomplicated structures (no tertiary structure) were expected to be largely unaffected by temperature, pH, metal ions, or various inhibitors.

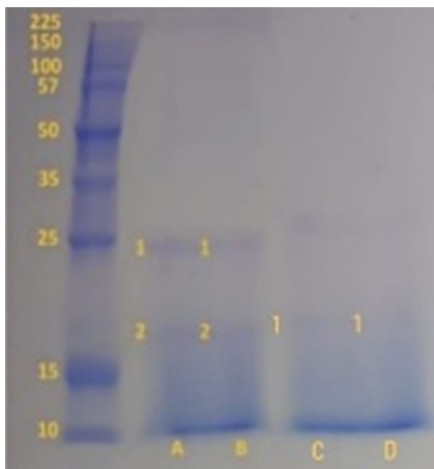

**Figure 4.** SDS-PAGE result of partial and purified bacteriocin. Lanes A and B: partially purified bacteriocin with ammonium sulphate has two protein bands. Lanes C and D: purified bacteriocin with gel filtration chromatography has one protein bands. 1 refers to 26.69 kDa; 2 refers to 17.15 kDa.

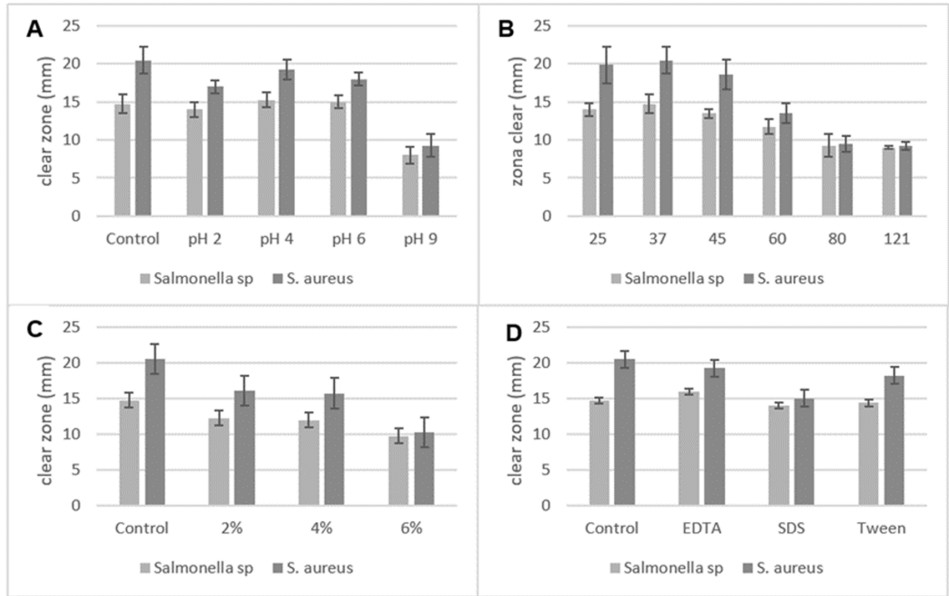

**Figure 5.** Partial characterization of the isolated antibacterial substance. (**A**) Effect of pH: the stability of the antibacterial substance at various pH values was measured after mixing the antibacterial substance with different pH values (2.0, 4.0, 6.0, and 9.0). (**B**) Effect of temperature: the antibacterial activity substance was heated at 25, 37, 45, 60, and 80 °C for 30 min, and at 121 °C for 15 min. (**C**) Antimicrobial activity with different salt concentrations (2%, 4%, 6%). (**D**) Antimicrobial activity with additional surfactant (EDTA, SDS, and Tween 80).

## 4. Discussion

The isolation process of lactobacilli was carried out in a dilution process up to 10-8, a multi-step dilution method aimed at reducing the colony density on the medium to facilitate the isolation process [26]. Isolates were grown using MRSA medium supplemented with 1% $CaCO_3$ and incubated at 37 °C for 48 h. The deMann, Rogosa, and Sharpe agar (MRSA) was a selective medium for growing lactic acid bacteria. When 1% $CaCO_3$ was added to the medium, lactobacillus colonies grew, as indicated by the presence of clear zones around the colonies [27]. The isolation resulted in five isolates: DSK 1, DSK 2, DSK 3, DSK 4, and DSK 5.

The five isolates were tested for antimicrobial activity and microscopic morphology to generate BAL candidates for bacteriocin production. The results indicated DSK 1 as the selected isolate with the highest antibacterial activity. Excessive acidic conditions cause lipopolysaccharide dissolution, pore formation that impairs energy synthesis and cell wall permeability, and ultimately leads to pathogen death. DSK 1 is a selected isolate to be further molecularly identified. A molecularly specific test was targeted to confirm dadih's specific strain of bacteria. Dadih products include LABs of various strains. Bacterial strains common in dadih include *Bifidobacterium*, *Lactobacillus*, *Lactococcus*, *Streptococcus*, and *Enterococcus* [28]. Molecularly, Lactobacillus was considered to determine specific strains. As a result of analysis, the lactic acid bacterium was found to be the *Lactobacillus pentosus* 124-2 strain.

Based on the results, the pure bacteriocin produced by gel filtration chromatography exhibited higher specific activity than that exhibited in the previous two methods. An increase in specific activity indicates an increase in bacteriocin purity and a constant value indicates that the bacteriocin is completely pure [29]. However, the yield values were too low and should be increased by optimizing production conditions, such as optimizing medium components and non-culturing conditions. Large-scale production of bacteriocins requires a purification process that achieves >50% yield at 90-fold purification speed [30]. Some bacteriocins have similar properties, allowing a comparison of different purification processes.

Antibacterial activity results showed that the bacteriocin *L. pentosus* was more resistant to the Gram-negative bacterium *Salmonella* sp. than to the Gram-positive *S. aureus*. Based on comparison, most bacteriocins produced by Lactobacillus species are only good against Gram-positive bacteria. Bacteriocin-derived *L. pentosus* was comparable to Nissin, which can only inhibit Gram-positive bacteria, and to *L. plantarum*'s bacteriocin, which can fight both Gram-positive and harmful bacteria, but it was unable to inhibit fungal growth. In contrast, some *L. pentosus* bacteriocins have been reported to be able to inhibit fungi such as Candida albicans, thus providing *L. pentosus* bacteriocins with the potential for broader applications [6]. However, bacteriocins have less antibacterial activity than antibiotics, but due to increasing antibiotic resistance, bacteriocins have yet to be developed and further studied.

Various factors affect the activity of bacteriocins, including temperature, pH, salinity, and surfactants. Characterization of changes in temperature, pH, concentration of NaCl, and surfactant type determined the activity potential of a bacteriocin as an antibacterial agent, its ability to meet the required antibiotic susceptibility pattern, and its inhibitory activity against pathogenic microorganisms. An evaluation test was developed [31]. The production of bacteriocins used as natural preservatives requires knowledge of their properties and viability under certain conditions in order to understand basic information about the function and specific properties of bacteriocins in their applications.

## 5. Conclusions

Overall, this study provides valuable insights into the potential use of the bacteriocin produced by Lactobacillus pentosus 124-2 as a natural preservative in the food industry. The purification process was successful in isolating a highly active and stable bacteriocin. The antibacterial activity of the bacteriocin was demonstrated against several Gram-positive bacteria, including *Staphylococcus aureus*. Importantly, the study also showed that increasing the specific activity of the bacteriocin during purification enhances its antibacterial activity, suggesting that further optimization of the purification process could improve the efficacy

of the bacteriocin. Moreover, the partial characterization of the bacteriocin revealed its stability over a wide range of pH and temperature, tolerance to salt, and stability in the presence of detergents. This suggests that the bacteriocin may have potential as a natural preservative in a variety of food products.

In conclusion, the bacteriocin produced by *Lactobacillus pentosus* 124-2 demonstrated promising characteristics that make it a potential candidate for use in food preservation. Further studies are needed to fully understand the mechanisms of action and potential applications of this bacteriocin in the food industry.

**Author Contributions:** Conceptualization, T.Y.; data curation, T.Y. and T.R.; formal analysis, A.R.P. and S.Z.; methodology, P.W.H.; software, A.R.P. and S.Z.; validation, Y.C.; investigation, T.R. and P.W.H.; writing—original draft, T.Y. and P.W.H.; writing—review and editing: T.Y., A.R.P., S.Z., T.R. and Y.C; project administration and funding acquisition, H.M. All authors have read and agreed to the published version of the manuscript.

**Funding:** This research and the APC were funded by the Universitas Padjadjaran, Indonesia, (Grant 2203/UN6.3.1/PT.00/2022) and by the Ministry of Research, Technology and Higher Education, Indonesia.

**Institutional Review Board Statement:** Not applicable.

**Informed Consent Statement:** Not applicable.

**Data Availability Statement:** The data presented in this study are available on request from the corresponding author.

**Acknowledgments:** T.Y. extends thanks to the Universitas Padjadjaran, Indonesia, for the funding support.

**Conflicts of Interest:** The authors declare no conflict of interest.

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
