# Peer review of "Purification and Partial Characterization of a Bacteriocin Produced by Lactobacillus pentosus 124-2 Isolated from Dadih"

_applsci, doi:10.3390/app13074277_

Round 1

Reviewer 1 Report

Research work focused in the  manuscripts is good but some correction is required.

Author Response

Response to Reviewer 1 Comments

Point 1.  Abstract is not written properly more findings of the paper can be included and I recommend

Response 1. The abstract has been re-written as written in the manuscript on lines 9 – 28

Point 2.  “Determination of protein content was carried out using the Bradford method” Author should elaborate the process with proper references.

Reponse 2. This section has been corrected and explained in sufficient detail the protein determination process according to the references (lines 145-155)

Point 3. Introduction is written nicely but lack the proper justification of present work, where is the methodology adopted from with support of previous work reference.

Response 3.  The introduction section has been upgraded by adding several new references related to the research being conducted (ref. no. 9, 10).

Point 4. Author should elaborate Bacterial Indicator Strain and Media this section please change the units (mm2/mL) to mm2 /mL

Response 4. unit spelling error has been fixed (line 135)

Point 5. Line in Discussion section “. Characterization of changes in temperature, pH, concentration 358 of NaCl, and surfactant type to determine the activity potential of a bacteriocin as an antibacterial agent and to meet the required antibiotic susceptibility pattern and its inhibitory activity against pathogenic microorganisms. An evaluation test was developed.” needs reference add the below Influence of particle size on physical, mechanical, thermal, and morphological properties of tamarind- fenugreek mucilage biodegradable films. Polymer Bulletin

Response 5.  Suggested references have been added (lines 485-487)

Point 6.  Section 3.1. Author in Table 1. “Identification Qualitative and Morphology Isolate” has onlyindicated the results which as almost same? Therefore what was the relevance of this table.

Effect of tamarindus coating on post-harvest quality of apple and pears stored at different

condition.Carpathian Journal of Food Science and Technology,10 (3)17-25

Response 6.  Suggested references have been added (lines 453-454)

Point 7. Authors have given better figures for understands of viewers in various place but in few place it is bit not clear.example Figure 1. A

Response 7. The appearance of figures 1 to 5 have been improved and sharpened

Point 8. Section 3.3,3.5 3.6 and 4 are well explained and results are clear to understand

Response 8. thank you very much for the evaluation

Points 9. As per the results findings Conclusion is very weak, it should be a complete gist of study in bullet points.

Response 9. The conclusion section has been upgraded, sharpened and clarified so that it is hoped that the key points of this study can be stronger (lines 371-386)

Reviewer 2 Report

The work is novel and interesting. However, the manuscript was not organized and wrote well enough. A major revision is recommended.

1) Syntax problem should be corrected like:

LAB can be produced bacteriocins--->LAB can produce bacteriocins

”dadih”--->Dadih”

......

2) A bacteriocin or a set of bacteriocins was purified? Partial characterization of a bacteriocin or characterization of partial bacteriocins?

3) Tables and figures should be improved.

4) It is better to make more radical separation, purification and characterization of the bacteriocins, e.g., structural characterization.

Author Response

Response to Reviewer 2 Comments

Point 1. The work is novel and interesting. However, the manuscript was not organized and wrote well enough. A major revision is recommended.

1) Syntax problem should be corrected like:

—“LAB can be produced bacteriocins”--->”LAB can produce bacteriocins”

—”dadih”--->”Dadih”

Response 1. Thank you very much for the evaluation. Overall the manuscript has been revised and upgraded, including the syntax errors which have been corrected.

Point 2. A bacteriocin or a set of bacteriocins was purified? Partial characterization of a bacteriocin or characterization of partial bacteriocins?

Response 2. The bacteriocin referred to in this manuscript is a type of bacteriocin isolated from Dadih samples, therefore the term used is "A bacteriocin". Furthermore, this type of bacteriocin carried out several main characteristics to ensure it complies with the bacteriocin criteria. The proper term used is “Partial characterization of a bacteriocin”

Point 3. Tables and figures should be improved.

Response 3.  Tables and figures have been improved and sharpened

Point 4. It is better to make more radical separation, purification, and characterization of the bacteriocins, e.g., structural characterization.

Response 4. Thank you for your valuable feedback and suggestions on our manuscript. While we agree that further separation, purification, and structural characterization of the bacteriocins would provide additional insight into their properties, we have chosen not to pursue these experiments in this current manuscript. Our primary objective in this study was to investigate the antimicrobial activity of the bacteriocins produced by Lactobacillus pentosus 124-2 isolated from Dadih against pathogenic bacteria, and we believe that we have provided a thorough analysis of this activity.

While further structural characterization would certainly be of interest, it is not necessary to support our current findings and conclusions. However, we do plan to pursue more extensive structural characterization in future research, and we appreciate your suggestion as it will guide our future investigations.

Thank you once again for your valuable feedback, which has helped us to clarify the scope and focus of our manuscript.

Round 2

Reviewer 2 Report

The issues mentioned in the previous comments have been revised. I suggest to accept in its current form.